# Integrating Remote Sensing and Ground-Based Data for Enhanced Spatial–Temporal Analysis of Heatwaves: A Machine Learning Approach

Thitimar Chongtaku [1], Attaphongse Taparugssanagorn [2,*], Hiroyuki Miyazaki [3] and Takuji W. Tsusaka [4]

1   Remote Sensing and Geographic Information Systems, Department of Information and Communications Technologies, School of Engineering and Technology, Asian Institute of Technology, P.O. Box 4, Klong Luang 12120, Pathum Thani, Thailand; st119790@ait.asia
2   Telecommunications, Department of Information and Communications Technologies, School of Engineering and Technology, Asian Institute of Technology, P.O. Box 4, Klong Luang 12120, Pathum Thani, Thailand
3   Center for Spatial Information Science, University of Tokyo, 5-1-5 Kashiwanoha, Kashiwa-shi 277-8568, Chiba, Japan; heromiya@csis.u-tokyo.ac.jp
4   Natural Resources Management, Department of Development and Sustainability, School of Environment, Resources and Development, Asian Institute of Technology, P.O. Box 4, Klong Luang 12120, Pathum Thani, Thailand
*   Correspondence: attaphongset@ait.asia

**Abstract:** In response to the urgent global threat posed by human-induced extreme climate hazards, heatwaves are still systematically under-reported and under-researched in Thailand. This region is confronting a significant rise in heat-related mortality, which has resulted in hundreds of deaths, underscoring a pressing issue that needs to be addressed. This research article is one of the first to present a solution for assessing heatwave dynamics, using machine learning (ML) algorithms and geospatial technologies in this country. It analyzes heatwave metrics like heatwave number (HWN), heatwave frequency (HWF), heatwave duration (HWD), heatwave magnitude (HWM), and heatwave amplitude (HWA), combining satellite-derived land surface temperature (LST) data with ground-based air temperature ($T_{air}$) observations from 1981 to 2019. The result reveals significant marked increases in both the frequency and intensity of daytime heatwaves in peri-urban areas, with the most pronounced changes being a 0.45-day/year in HWN, a 2.00-day/year in HWF, and a 0.27-day/year in HWD. This trend is notably less pronounced in urban areas. Conversely, rural regions are experiencing a significant escalation in nighttime heatwaves, with increases of 0.39 days/year in HWN, 1.44 days/year in HWF, and 0.14 days/year in HWD. Correlation analysis ($p < 0.05$) reveals spatial heterogeneity in heatwave dynamics, with robust daytime correlations between $T_{air}$ and LST in rural (HWN, HWF, HWD, $r > 0.90$) and peri-urban (HWM, HWA, $r > 0.65$) regions. This study emphasizes the importance of considering microclimatic variations in heatwave analysis, offering insights for targeted intervention strategies. It demonstrates how enhancing remote sensing with ML can facilitate the spatial–temporal analysis of heatwaves across diverse environments. This approach identifies critical risk areas in Thailand, guiding resilience efforts and serving as a model for managing similar microclimates, extending the applicability of this study. Overall, the study provides policymakers and stakeholders with potent tools for climate action and effective heatwave management. Furthermore, this research contributes to mitigating the impacts of extreme climate events, promoting resilience, and fostering environmental sustainability.

**Keywords:** heatwaves; data gap-filling; remote sensing; satellite data; air temperature; machine learning; random forest; geospatial artificial intelligence; natural hazard; Thailand

## 1. Introduction

Heatwaves, acknowledged as silent killers, are defined as prolonged periods of excessively hot weather, which may also be accompanied by high humidity and are considered

among the deadliest climatic hazards [1]. Over the initial two decades of the 21st century, a multitude of severe heatwaves had a profound impact on human health, agriculture, water resources, energy demand, regional economies, and forest ecosystems [2]. Extreme events of this phenomenon have far-reaching and detrimental consequences at global, national, and local levels. They impose significant impacts on societies, including increased morbidity and mortality rates, overwhelming healthcare systems, diminished agricultural and ecosystem productivity, and substantial economic losses [3–12]. Furthermore, heatwaves exacerbate the vulnerability of built environments, leading to increased energy demand for cooling and a heightened risk of infrastructure failure. Climate change, recognized as a major influencer of global damage, is exacerbating heatwaves, making them hotter, longer-lasting, and more frequent, thereby amplifying their impacts on people, property, communities, and the environment [13].

In particular, urban environments are significantly more susceptible to elevated temperatures due to the 'urban heat island (UHI)' effect [14,15]. This susceptibility is further exacerbated by multiple factors, such as limited green spaces, high population density, compromised air quality, restricted availability of cooling resources, and a high concentration of buildings [16]. However, vulnerability to heatwave effects extends beyond the city center, indicating that suburban residents are also at risk. These areas, characterized by their heightened sensitivity and limited adaptive capacities, experience particularly high risks of heat-wave-related mortality [17]. Therefore, the observed pattern of extreme heat phenomena not only underscores a critical gap in our current understanding but also highlights the urgent necessity for in-depth studies with a precise focus on diverse environments to ensure a comprehensive assessment of effects across varied areas.

Heatwaves are a serious threat in many countries throughout the world. Between 1998 and 2013, extreme heat events were responsible for over 100,000 deaths across 164 locations in 36 countries, a number that rose to 166,000 by 2017 [18]. The unprecedented heatwave of 2019 caused roughly 5000 deaths in Europe, Asia, and Australia amid record-breaking temperatures. The summer of 2022 saw another series of record highs, impacting South Asia, North America, Europe, and China, and resulting in an estimated 15,000 heat-related fatalities in Europe alone [19,20]. Moreover, projections from the Eurostat mortality database indicate a troubling rise in heatwave-related deaths globally, with estimates predicting increases to 68,000 by 2030, 94,000 by 2040, and 120,000 by 2050 [21]. Predictive models forecast a surge in the frequency of these extreme temperature events, expected to be up to seven times more frequent in the next three decades [22,23]. Alarmingly, heatwaves have intensified rapidly in some Asian regions, particularly in South and Southeast Asia, where the number of heatwave days has increased by 4.2 days per decade, compared to the global average of 2.26 days per decade [24–27]. Thus, the increasing occurrence of high-magnitude and impact heatwaves has raised concerns worldwide and has attracted increasing interest in this issue among researchers over the past decade [28]. For that reason, these observations expose a critical gap in the need for thorough research on heatwave phenomena, especially in the context of ongoing climate change.

Southeast Asia, highly susceptible to the detrimental impacts of extreme heat, faces significant challenges [29]. The increase in heatwave amplitude has shown a linear growth in relation to global warming levels, with distinct regional differences between the Maritime Continent and the Indochina Peninsula due to their differing heat content in lower atmospheric boundaries. The trend towards more frequent, prolonged, and intense heatwaves is projected to continue, exacerbating public health challenges and underscoring the imperative for comprehensive adaptive strategies to mitigate the impacts of these devastating climate phenomena, particularly in countries like Thailand [30]. Thailand itself has experienced fluctuations in climate, particularly in rainfall patterns and temperature [31]. With its relatively underdeveloped public health infrastructure and highly vulnerable populations [32], the region is confronting increasingly severe climate-related issues. Notable increases in temperatures in recent years have led to alarming health repercussions, including a significant rise in heat-related deaths [33]. Between 2015 and 2018, heatwaves in

Thailand led to 158 deaths, primarily in the northern and central provinces, where extreme temperatures are directly linked to higher mortality rates [34]. Concurrently, there has been a rise in respiratory and cardiovascular diseases, now among the top five causes of death and disability in the country [35]. Despite these challenges, current heatwave management practices remain underdeveloped. Ref. [36] recommends an integrated approach to urban planning and design to mitigate heat stress. Critical measures include the development of green open spaces to reduce urban heat island effects and the implementation of green building codes, which advocate for features like rooftop gardens and heat-reducing materials. Furthermore, establishing heat early warning systems and collaborative emergency response plans is vital for enhancing urban resilience against heat stress, particularly in cities like Bangkok.

Satellite remote sensing offers significant promise for the precise study and detection of heatwaves from a broad perspective, aiding in the comprehension of heatwave patterns [37]. Defined by its high spatial resolution and temporal frequency, this approach allows for extensive analyses, yielding regular and long-term data crucial for grasping the behavior and effects of heatwaves. This technique overcomes the limitations posed by unavailable or poorly distributed ground station networks [38,39]. Previous studies [16,40–46] have demonstrated the capability of multitemporal remote sensing data from several satellites to analyze, map, and monitor the spatial and temporal dynamics of heatwaves.

Furthermore, studies by [47,48] have shown that thermal satellite-derived LST is a crucial parameter for identifying heatwaves and understanding the consequences of extreme heat. Additionally, LST plays a vital role as a key input in the study of land surface water and energy budgets at both local and global scales [49]. LST, recognized as one of the essential climate variables (ECVs) by the World Meteorological Organization (WMO), is a key indicator for both climate change and land surface processes. This is due to the heat exchange between the land surface and the near-surface atmosphere, making the dynamics in air temperature and LST consistent [50,51]. Various satellites with Thermal Infrared (TIR) airborne sensors, including the Advanced Very-High-Resolution Radiometer (AVHRR), the Moderate Resolution Imaging Spectroradiometer (MODIS), and LANDSAT, have been used for this purpose [39,52–54]. Among these, data from MODIS have been pivotal for the retrieval of LST dynamics and trends, providing the longest consistent time series covering vast global regions [55]. Its global radiometric resolution and dynamic ranges, along with accurate calibration in multiple TIR bands, are well designed [56]. Furthermore, MODIS offers the advantage of revisiting the same area four times daily, ensuring detailed and timely data collection. However, the process of acquiring LST data from TIR observations often encounters obstacles due to the presence of cloud cover, which introduces widespread data gaps, affecting statistically more than 60% of the global extent [57]. This issue poses substantial challenges in analyzing the spatial and temporal variations of LST.

### 1.1. Research Gaps

To date, in response to these challenges, an extensive array of research initiatives have been undertaken with the aim of developing sophisticated techniques for the reconstruction of missing data within LST datasets [49,58–62]. A number of methods have been developed, which can be generally divided into three types according to the sources of reference information: (1) spatial information, (2) multitemporal observations, and (3) other complementary data, for example, from ground meteorological stations [59]. However, techniques that rely on comparing cloudy-sky pixels to nearby clear-sky ones are effective only in images with minimal cloudiness [62,63]. Although meteorological station data can provide the temperature of the point where the station is located, accurate and precise heat wave analysis requires continuous temperature distribution [41,64]. To address these gaps, an improved method for reconstructing LST in cloud-covered areas has been proposed. This method employs a linking model that integrates MODIS-LST with other surface variables, such as surface topography and land cover conditions, through a machine learning algorithm known as random forest (RF) regression [49]. Owing to its robust predictive

performance and capability to process complex, non-linear data, this approach not only enables accurate predictions of various extreme climate phenomena but also facilitates a comprehensive evaluation of the significance of different temporal and spatial features in predicting LST.

### 1.2. Our Contributions

Heatwaves denote prolonged periods of exceptionally hot weather, often coupled with high humidity, exerting severe repercussions on human health and the environment [1]. Despite extensive research endeavors [3–12], including investigations into urban areas vulnerable to the 'urban heat island (UHI)' effect [14,15], our current grasp of heatwave effects remains incomplete. This gap underscores the need for comprehensive studies focusing on diverse environments to ensure a thorough assessment of heatwave impacts across varied areas.

Our contribution addresses this critical gap by introducing a novel approach for understanding heatwave dynamics, particularly in data-deficient regions vulnerable to severe heatwave events in Thailand. We aim to elucidate the complex temperature dynamics in such regions, encompassing urban (Bangkok), suburban (Pathum Thani), and rural (Saraburi) locations. Leveraging LST as a vital indicator for heatwaves, we propose an innovative, integrated RF model that combines satellite-derived LST data with air temperatures and spatial and temporal features. Unlike previous methods [16,37,40–46] reliant solely on remote sensing data or ground-based observations, our model bridges the gap between the two sources, enhancing the accuracy and reliability of heatwave predictions.

Satellite remote sensing holds promise for studying heatwaves due to its high spatial resolution and temporal frequency [47,48]. However, cloud cover often obstructs LST data acquisition, leading to widespread data gaps [57]. Our proposed method overcomes this challenge by integrating MODIS-LST with other surface variables, such as surface topography and land cover conditions, through RF regression. This integration enhances the predictive performance of our model, enabling accurate predictions of various extreme climate phenomena and facilitating a comprehensive evaluation of spatial–temporal heatwave patterns across diverse environments. Thus, our study represents a significant advancement in understanding heatwave dynamics and provides valuable insights for effective heatwave risk management and climate resilience strategies.

Overall, our findings underscore the significant role of earth observation and machine learning in shaping sustainable climate strategies. Notably, recent advancements in machine and deep learning approaches have markedly enhanced prediction accuracy and research outcomes over traditional methods. These developments are pivotal in informing governmental policies and guiding decision-making processes, leading to improved resilience in both urban and non-urban environments against climate extremes and driving efforts toward a more sustainable future.

## 2. Materials and Methods

This section describes the geographical context, data collection methods, and analytical techniques, providing insights into the scientific framework that underpins the investigation of heatwave dynamics in Thailand's central region. It begins with a description of the study area, followed by an exploration of the data sources used in the analysis. The section then outlines the methodologies employed for the spatial–temporal quantification of heatwaves and the spatial correlation analysis, highlighting the novel approaches adopted to achieve the research objectives.

### 2.1. Study Area Description

In this study, we focus on urban, peri-urban, and rural regions within the central region of Thailand as depicted in Figure 1. These classifications adhere to national standards and are based on criteria such as administrative divisions, demographic profiles, primary land use, and prevalent occupations as cited in [65,66]. Therefore, the selected study areas

are systematically categorized into urban, peri-urban, and rural types according to these variables. Additionally, these areas are situated within a tropical monsoon climate zone, characterized by three distinct seasons: summer (mid-February to mid-April), rainy (mid-April to mid-October), and winter (mid-October to mid-February). The selected regions are as follows:

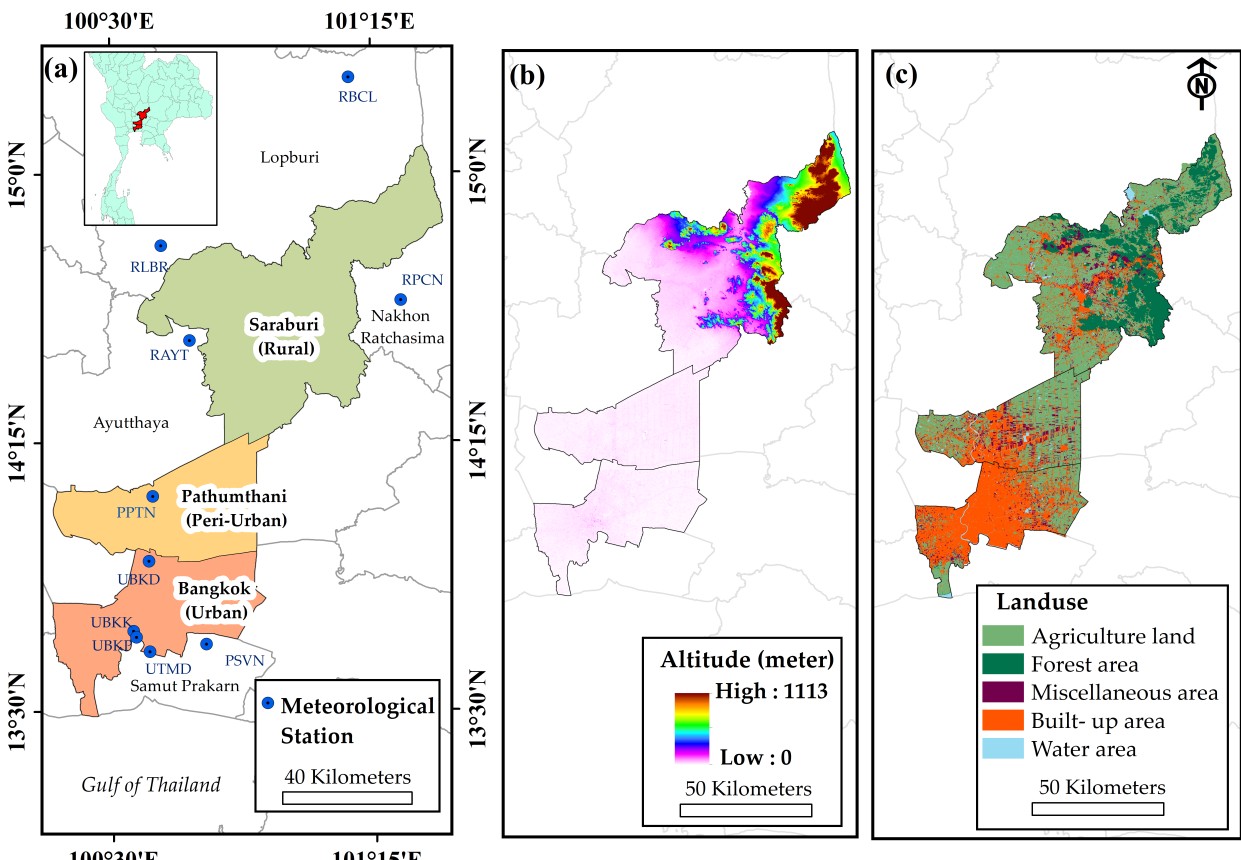

**Figure 1.** Location of the study area in Thailand and spatial distribution of the meteorological stations in dotted print (**a**), altitude (**b**), and land use (**c**).

- Urban—Bangkok province is located at latitude 13°38′ N and longitude 100°54′ E. This area is flat and low-lying, with altitudes ranging from 1.50 m to 2.0 m. It occupies 1600 km$^2$ and consists of 50 districts with a registered population of 5.52 million, making it the most highly inhabited city in Thailand. The built-up area covers 67.36%, while agricultural land constitutes only 0.16% of the total area. The annual mean air temperature ranges from 28.0 °C to 30 °C, with the average maximum air temperature in April ranging from 33.0 °C to 38.0 °C, and the average minimum air temperature in January ranging from 16.0 °C to 25.0 °C.
- Peri-urban—Pathum Thani province is situated at latitude 14°01′ N and longitude 100°32′ E, with an average altitude of 2.30 m. It covers an area of 1519 km$^2$ and comprises 7 districts with a registered population of 1.19 million. The annual mean air temperature ranges from 28 °C to 30 °C, with the average maximum air temperature in April ranging from 32 °C to 34 °C, and the average minimum air temperature in December ranging from 24.0 °C to 26.0 °C. This area is characterized by 29.79% of the land serving as city outskirts, accommodating dense dwellings, industrial estates, and being devoid of forest areas.
- Rural—Saraburi province is situated at latitude 14°31′ N and longitude 100°54′ E, with the landscape ranging from 2.0 m to 10.0 m in elevation, covering an area of 3576 km$^2$. This region includes 13 districts and has a population of 643,963 registered



residents. The average annual air temperature is 28.2 °C, with highs of 31.4 °C in April and lows of 23.6 °C in January. Approximately 78% of the land consists of agricultural and forested areas.

### *2.2. Data Sources*

### 2.2.1. Ground-Observed Air Temperature

Ground-based air temperature data, including daily maximum air temperature ($T_{max}$) and daily minimum air temperature ($T_{min}$), from 1981 to 2019, were acquired from the meteorological stations operated by the Thai meteorological department (TMD). These observations are measured at 2.0 m above the ground according to the world meteorological organization (WMO) standard [67]. In this research, temperature data from 10 meteorological stations are utilized as depicted by the dotted points in Figure 1a. These stations are classified into urban, peri-urban, and rural categories based on their geographical locations as outlined in Table 1. As there is no ground station observed in Sara Buri province, which represents the rural area, this study designates the meteorological stations in Ayutthaya, Lopburi, and Nakhon Ratchasima to serve as proxies for this inactive area.

**Table 1.** Meteorological station details and available data time period.

| Station | ID | Station Name | Province | Altitude (m) | Type of Area | Time Period (Years) |
|---|---|---|---|---|---|---|
| 1 | UBKP | Bangkok Port (Khlong Toei) | Bangkok | 1 | Urban | 1994–2019 (26) |
| 2 | UBKK | Bangkok (Queen Sirikit National Convention Center) | Bangkok | 4 | Urban | 1981–2019 (39) |
| 3 | UTMD | Thai Meteorological Department (Bang Na) | Bangkok | 3 | Urban | 1981–2019 (39) |
| 4 | UBKD | Don Muang Airport | Bangkok | 5 | Urban | 1981–2019 (39) |
| 5 | PSVN | Suvarnabhumi Airport | Samut Prakan | 2 | Peri-urban | 2008–2019 (12) |
| 6 | PPTN | Pathum Thani Agrometeorological Station | Pathum Thani | 9 | Peri-urban | 1998–2019 (21) |
| 7 | RAYT | Ayutthaya Meteorological Station | Ayutthaya | 12 | Rural | 1993–2019 (27) |
| 8 | RLBR | Lopburi Meteorological Station | Lopburi | 20 | Rural | 1981–2019 (39) |
| 9 | RBCL | Bua Chum Meteorological Station | Lopburi | 54 | Rural | 1981–2019 (39) |
| 10 | RPCN | Pak Chong Meteorological Station | Nakhon Ratchasima | 422 | Rural | 1981–2019 (39) |

### 2.2.2. Remotely Sensed Land Surface Temperature

The MODIS sensors, aboard NASA's Terra and Aqua spacecraft, launched in 1999 and 2002 respectively, play a pivotal role in global studies of Earth's surface, atmosphere, cryosphere, and ocean processes [68]. These instruments capture data across 36 spectral channels from 3 to 15 μm, ranging from visible to infrared wavelengths. Data are quantized to 12 bits and offer a spatial resolution of approximately 1 km at nadir, with overpass times at approximately 10:30 and 22:30 local solar time (Terra) and 13:30 and 01:30 (Aqua) of any location on Earth every 1–2 days [69]. Specifically, LST data are generated from thermal infrared bands 31 and 32 (at 11 and 12 μm) using a generalized split-window (GSW) algorithm with an accuracy around 2.0 K [70]. This physics-based algorithm, developed to address challenges such as atmospheric transmission, path radiance, downward thermal irradiance, and solar diffuse irradiance, simultaneously retrieves surface band-averaged

emissivities and temperatures. It efficiently processes temperature measurements from day/night pairs of MODIS data [38,56,71].

Two specific MODIS-LST products (version 6.1), which have been enhanced through various calibration changes, were incorporated into our analysis. Both products are sourced from tile h27v07: the Terra daily LST (MOD11A1), spanning from 1 January 2000 to 31 December 2019, and the Aqua daily LST (MYD11A1), covering the period from 1 January 2002 to 31 December 2019. We selected these time frames to provide a comprehensive view of temperature anomalies over nearly two decades. For data extraction and analysis, we utilized the $A\rho\rho$EEARS application, which is accessible at https://appeears.earthdatacloud. nasa.gov/. This tool facilitates efficient access to geospatial data, enabling the point extraction of MODIS-LST and supplementary data.

### 2.2.3. Elevation and Land Use/Land Cover Data

The study area's elevation was determined using the publicly accessible shuttle radar topographic mission (SRTM) digital elevation model (DEM) from the USGS earth explorer platform. The DEM has a spatial resolution of 90 m. Subsequently, the data at this resolution were averaged to achieve a finer 10 m resolution as demonstrated in Figure 1b. Additionally, land use/land cover data from the land development department (LDD) of Thailand were incorporated. These raster-format data cover specific regions, including Bangkok, Pathum Thani, Samut Prakan, Ayutthaya, Lopburi, and Nakhon Ratchasima, spanning the years 2000–2019. The data are classified according to the Level 1 LDD standard classification criteria, comprising five classes as illustrated in Figure 1c: built-up area (U), agricultural land (A), forest area (F), water body (W), and miscellaneous land use (M). This classification aids in comprehending regional land use patterns and their potential impact on local temperatures.

### 2.3. Methods

This section outlines the methodology used to examine the spatial–temporal quantification of heatwaves and to measure the degree of spatial heterogeneity. It aims to present the potential of LST-based satellite data for understanding and monitoring heatwave events in different regions of Thailand. Initially, the approach encompasses data collection from diverse sources. Then, it addresses missing data through imputation due to technical malfunctions. Subsequently, the methodology involves identifying heatwave metrics. Finally, it visualizes the spatial–temporal patterns of heatwaves to align detected heatwaves $T_{air}$ with LST, thus presenting a comprehensive processing sequence as illustrated in Figure 2.

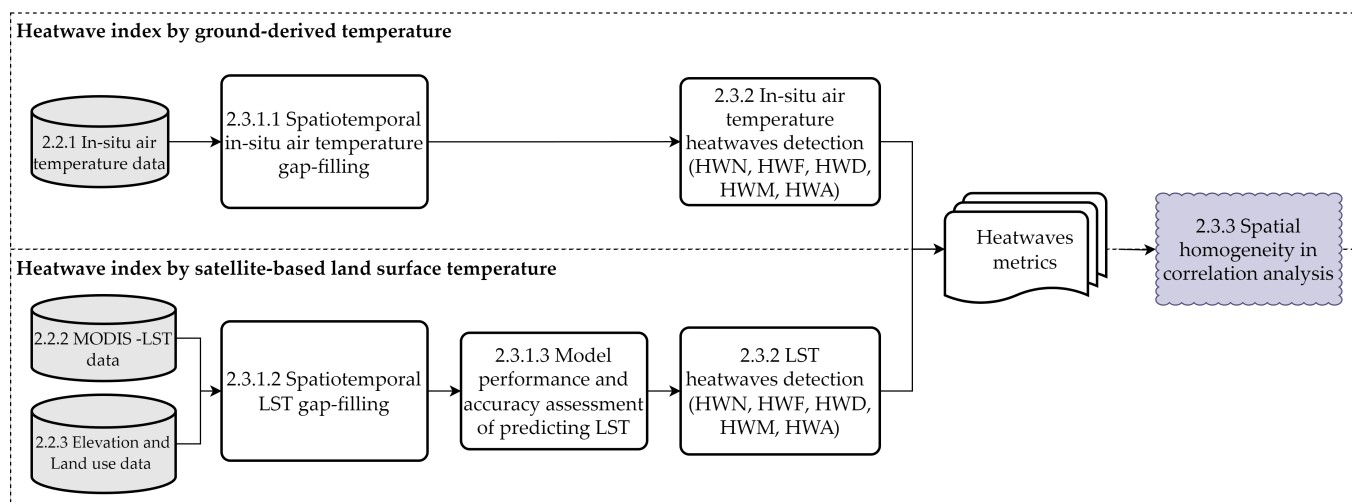

**Figure 2.** Data utilization, methodology, and study findings in relation to the organization of sections within this research paper.

### 2.3.1. Data Gap-Filling

In the preprocessing phase, addressing the challenge of missing values is vital to ensure the integrity and precision of climate-data-driven predictions. The dataset undergoes optimization through the systematic removal or statistical imputation of these missing values, rendering it more suitable for subsequent predictive modeling or cluster analysis functions.

Spatial–Temporal Ground-Observed Air Temperature Gap-Filling

In the event of malfunctioning at the meteorological station, the $T_{air}$ dataset encounters partial incompleteness. Recognizing the missing data adherence to a pattern of missing at random (MAR), ref. [72] suggests employing straightforward imputation methods to mitigate these gaps. When the extent of missing data remains below 5 percent, a mean substitution approach is advocated. This entails replacing missing values with the mean computed from observed values spanning multiple years.

However, if the percentage of missing data exceeds 5% or spans over a month, a regression model is employed.

Regression models, whether utilizing a single closest neighbor station [73] or multiple nearby stations [74–76], have proven effective in estimating daily weather observations [74,77–79]. The imputation processes, which are applied to both $T_{max}$ and $T_{min}$, are represented by

$$E(Y|X) = \beta_0 + \beta_1 X, \tag{1}$$

where $Y$ is the predicted missing air temperature, $X$ is the reference data from a neighboring station, $E(Y|X)$ is the expected value of $Y$ given the values of $X$, $\beta_0$ is the intercept, and $\beta_1$ is the slope.

Spatial–Temporal Satellite-Based Land Surface Temperature Gap-Filling

Addressing gaps in daily LST data derived from MODIS platforms poses a significant challenge, necessitating innovative solutions for data reconstruction and prediction. Recent studies have explored various methods, including reconstructing LST from satellite datasets, forecasting daily LST using time series data, and estimating subpixel LST by fusing multisource data [80–83]. Although machine learning techniques have been introduced to LST retrieval, their application remains limited [84]. To bridge this gap, we employ the RF machine learning algorithm, celebrated for its exceptional predictive ability and adaptability to nonlinear data [85]. The RF methodology of constructing multiple decision trees provides a robust framework for classification and regression tasks, making it particularly effective for filling gaps in LST data caused by cloud cover or other disruptions.

The RF algorithm is adept at managing high-dimensional data and unraveling complex relationships, which is crucial for accurately predicting missing LST values. It is noted that the RF performance is enhanced when there is a strong correlation between the target and reference variables, a principle that holds true in the analysis of intricate environmental datasets [86].

By leveraging RF, we can address missing data without relying on standard scalar functions. Although these functions are useful for normalization and standardization, they can potentially introduce bias. Instead, our approach focuses on utilizing the Python-based scikit-learn module [87] for efficient gap-filling, ensuring that the integrity and natural variability of the LST dataset are maintained. This methodology not only facilitates the prediction of missing LST values with minimal bias but also underscores the potential of machine learning in advancing the understanding and monitoring of extreme climate events [88–91].

In RF, the prediction is made by aggregating predictions from multiple decision trees. The prediction $\hat{y}$ for a given sample $x$ is calculated by

$$\hat{y} = \frac{1}{N} \sum_{i=1}^{N} f_i(x), \tag{2}$$

where $N$ is the number of decision trees in the forest, and $f_i(x)$ is the prediction of the $i$-th decision tree.

Model Performance and Accuracy Assessment of Predicting Land Surface Temperature

A comprehensive set of statistical measures is computed on the test set to assess the accuracy and performance of the predictive model for LST. These measures include important metrics like the coefficient of determination ($R^2$), root mean square error ($RMSE$), minimum and maximum confidence intervals, mean absolute error ($MAE$), and mean bias error ($MBE$) as

$$R^2 = 1 - \frac{\sum_{i=1}^{n} (\hat{y}_i - \underline{y})^2}{\sum_{i=1}^{n} (y_i - \underline{y})^2}, \tag{3}$$

$$RMSE = \left[ \sum_{i=1}^{n} \frac{(y_i - \hat{y}_i)^2}{n} \right]^{1/2}, \tag{4}$$

$$MAE = \frac{1}{n} \sum_{i=1}^{n} |y_i - \hat{y}_i|, \tag{5}$$

$$MBE = \frac{1}{n} \sum_{i=1}^{n} (y_i - \hat{y}_i), \tag{6}$$

where $N$ is the number of records in validation datasets used in this study, $\hat{y}_i$ is the estimated variable, $y_i$ is the observed variable, $\underline{y}$ is the mean of all the values, and the confidence level is set to 0.95.

For the assessment, the calibration phase utilizes variable data spanning from 2000 to 2017, while the subsequent validation phase involves data from 2017 to 2019, following an 80% train and 20% test split. The $R^2$ value, ranging from 0 to 1.0, provides a valuable indication of the predictive model's accuracy in estimating outcomes. The RMSE is effectively utilized to evaluate biases in both mean and spatial variance, whereas the MAE serves as a reliable measure of error magnitude, with lower values demonstrating superior performance. Additionally, the MBE offers insightful observations regarding the direction of error bias, with a value of zero denoting an unbiased estimation by the model [92,93].

2.3.2. Heatwave Definition and Its Metric Detection

To date, a universally accepted definition of heatwaves remains inconsistent. Previous investigations have adopted a variety of methodologies, leading to variations influenced by meteorological conditions, socio-demographic attributes, acclimatization processes, and geographic factors [94]. Nonetheless, a common metric employed by numerous investigators involves the 90th percentile of daily maximum and minimum temperatures, designated as CTX90pct and CTN90pct, respectively. This methodology has been adopted in various studies [95–97], establishing a standardized and globally relevant framework for quantifying heatwave characteristics such as frequency, duration, and intensity. Such a standardized approach facilitates the derivation of threshold values that are instrumental across a multitude of geographical locales and sectors of impact [38,98], proving particularly valuable in tropical regions, including Thailand [99].

In this study, the detection and measurement of $T_{air}$ and LST heatwaves are conducted using established methods referenced in [95,100,101] and summarized in Table 2. A heatwave is defined as a period when temperatures exceed a certain threshold for at least three consecutive days, effectively identifying sustained extreme temperature events and reflecting temporal variations. This methodology, supported by findings from [30], is particularly well suited for Southeast Asia due to its complex land–sea configuration and diverse topographies, offering a nuanced approach to detecting regional heatwave patterns. We observe both daytime and nighttime heatwaves as widely used in [102–106]. Daytime heatwaves, essentially defined by high daytime maximum temperatures ($T_{max}$), are accompanied by increased downward shortwave radiation under clear skies with re-

duced cloud cover and moisture, as well as lower humidity. These hot and dry daytime conditions can lead to potential impacts such as wildfires, water deficits, reduced crop yields, and increased human health risks [107]. In contrast, nighttime heatwaves, often measured by high nightly minimum temperatures ($T_{\min}$), typically occur under moist conditions characterized by increased cloud fraction, humidity, and long-wave radiation at the surface. These conditions significantly affect human comfort and inhibit recovery from the heat experienced during the daytime, thus increasing the threat to human health from high-temperature weather. More hazardous conditions emerge when extreme daytime temperatures are combined with warm nighttime conditions for consecutive days, creating compound heatwaves [106].

**Table 2.** Heatwaves indices used in the analysis.

| Index | Abbreviation | Definition | Unit | Reference |
|---|---|---|---|---|
| Heatwave number | HWN | The total number of individual heatwaves detected occurs when temperatures exceed the 90th percentile of a given temperature for at least three consecutive days | events | [95,100,101] |
| Heatwave frequency | HWF | The total number of days that contribute to heatwaves | days | [95,100,101] |
| Heatwave duration | HWD | The length in days of the longest heatwave | days | [95,100,101] |
| Heatwave magnitude | HWM | The average of mean daily temperature throughout the duration of heatwave | °C | [95,100,101] |
| Heatwave amplitude | HWA | The peak daily value in the hottest heatwave of the highest HWM | °C | [95,100,101] |

To define daytime heatwave events, this study uses daily maximum air temperatures ($T_{\max}$) from MODIS-MOD11A1 day and MODIS-MYD11A1 day datasets. Nighttime events are similarly identified using daily minimum air temperatures ($T_{\min}$) from MODIS-MOD11A1 night and MYD11A1 night. Heatwaves are determined using a criterion based on the 90th percentile of the calendar day temperatures.

This threshold captures the annual variation in extreme heat, with a unique percentile calculated for each day of the study period. As a result, this approach encompasses all heatwave events occurring from the start to the end of the period of interest. The CTX90pct method is utilized for $T_{\max}$ and LST day (MOD11A1 day, MYD11A1 day), while the CTN90pct method is employed for $T_{\min}$ and LST night (MOD11A1 night, MYD11A1 night), the 90th percentile threshold ($T_{90}$) for a given set of temperature data $T$ is calculated as

$$T_{90} = \text{Percentile}(T, 90), \tag{7}$$

where $\text{Percentile}(T, 90)$ represents the temperature value below which 90% of the observations fall in the dataset ($T$).

2.3.3. Spatial Homogeneity in Correlation analysis

The point-to-pixel analysis method is employed to match the series of heatwave metrics data, calculated based on $T_{\text{air}}$, with the corresponding pixels of gridded LST data. Only pixels that have at least one available and used a ground-based gauge for the calculation are included [108,109]. Given the extensive research area and the time required for retrieving daily MODIS data, a sampling design with a buffer zone at the confluence of the regular 5 km × 5 km latitude and longitude grid is adopted for validating $T_{\text{air}}$ and LST heatwaves. The use of available grids is because heatwaves may not occur at every grid, considering the restriction of three consecutive days exceeding the threshold temperature in the heatwave definition [30]. Therefore, this produces a total of 248 valid grids encompassing three provinces of study.

To establish the spatial correlation between the pixel values of $T_{\text{air}}$ and the corresponding pixels in the detected heatwave from the LST dataset, the pixel-wise correlation is determined using the Pearson correlation coefficient ($r$) throughout the research period.

Furthermore, the spatial–temporal relationship between a meteorological station grid and the four nearest LST grids is determined using the median value of the four Pearson's $r$ coefficients. Notably, the $T_{max}$ is cross-validated with daytime LST (MOD11A1 (day) and MYD11A1 (day)), whereas the $T_{min}$ is evaluated with nighttime LST (MOD11A1 (night) and MYD11A1 (night)). Spatial homogeneity in the correlation analysis can be defined as

$$r = \frac{\sum_{i=1}^{n}(x_i - \bar{x})(y_i - \bar{y})}{\sqrt{\sum_{i=1}^{n}(x_i - \bar{x})^2}\sqrt{\sum_{i=1}^{n}(y_i - \bar{y})^2}}, \tag{8}$$

where $r$ is the correlation coefficient, $x_i$ is the values of the x-variable in a sample $i$, $\bar{x}$ is the mean of the values of the x-variable, $y_i$ is the values of the y-variable in a sample $i$, and $\bar{y}$ is the mean of the values of the y-variable.

2.3.4. Software Tools for Data Manipulation, Analysis, and Visualization

For the manipulation, analysis, and visualization of data in this study, we utilized a comprehensive array of software tools. These included ArcMap (10.7, licensed by Asian Institute of Technology) and QGIS Desktop (3.4.12) for spatial data analysis and mapping, as well as R Studio and Python for statistical analysis, data processing, and graphical representation. Microsoft Excel was also employed in conjunction with R Studio and Python for detailed statistical analyses.

Regarding specific methodologies, the RF regression model was implemented using the **sklearn.ensemble.RandomForestRegressor** from the sklearn package [110], ensuring efficient gap-filling and preservation of the integrity and natural variability of the LST dataset. For the MK test, we employed the Python package **pyMannKendall** [111], which offers a robust and efficient means of calculating the test statistic, ensuring reliable statistical results.

**3. Results**

*3.1. Predicting Land Surface Temperature to Fill in Missing Satellite Data and Variable Importance*

To assess the effectiveness of RF models in predicting LST to fill the gaps of missing LST data, we meticulously designed an experiment incorporating both temporal and spatial variables. Our RF models to predict LST missing data include both temporal (day of the year (DOY) and year) indicators and spatial variables ($T_{max}$, $T_{min}$, elevation, built-up land, agricultural land, forest land, water bodies, and miscellaneous land), allowing for an evaluation of feature importance specifically for predicted LST.

To implement this approach, we systematically divided the study area into a grid with 1 km intervals around each of the 10 meteorological stations. Subsequently, we selected four points along cardinal directions to generate predicted LST models for both daytime and nighttime. Following the creation of these LST prediction models, we further divided the area into a grid with 5 km intervals, covering all seven provinces with weather stations. This resulted in 11 training samples and 1250 testing samples. Notably, only 248 points within this grid overlapped with the targeted study area encompassing Bangkok, Pathum Thani, and Saraburi provinces. This design ensures a rigorous evaluation of the RF models' predictive capabilities, accounting for both temporal and spatial factors, while also considering the specific features of the targeted study areas.

According to the RF results, a concise overview of the model's feature importance for LST modeling is presented in Figure 3. The factor DOY shows the highest explanatory rates for daytime LST as in MOD11A1_day (Figure 3a) and MYD11A1_day (Figure 3b), whereas the $T_{min}$ affects most LST night as in MOD11A1_night (Figure 3c) and MYD11A1_night (Figure 3d). In predicting LST day, $T_{max}$ and $T_{min}$ rank second, followed by year, elevation, and land use. For MOD11A1 data, the water body from the land use factor is essential, while the built-up area is vital for MYD11A1 data. Later, in the case of MOD11A1 and MYD11A1, $T_{min}$ emerges as the second most influential factor for predicting nighttime LST, following the elevation of the land and the extent of built-up areas.

A summary of the LST model statistical measurement between the observed LST and predicted LST is given in Table 3. The result demonstrates that MOD11A1 night has the greatest satisfied RMSE (2.09 °C) and the best linear relationship ($R^2 = 0.64$), followed by MYD11A1 night ($R^2 = 0.48$), and MOD11A1 day ($R^2 = 0.45$). On the other hand, MYD11A1 day is an unsatisfactory model due to the highest RMSE (5.02 °C) and lowest $R^2$ (0.29).

These results confirm the ability of the RF machine learning algorithm, particularly the MOD11A1 night model, in estimating LST. Conversely, the other predicted LST models indicate a tendency to overestimate the data.

**Table 3.** Evaluation of calibrated and validated land surface temperature predictions.

| Statistical Measures | MOD11A1_day (C/V) | MYD11A1_day (C/V) | MOD11A1_night (C/V) | MYD11A1_night (C/V) |
|---|---|---|---|---|
| $R^2$ | 0.50/0.45 | 0.51/0.29 | 0.55/0.64 | 0.64/0.48 |
| RMSE | 2.64/3.79 | 2.79/5.02 | 2.03/2.09 | 1.79/2.57 |
| Min Interval | 2.52/3.79 | 2.66/5.02 | 1.92/2.08 | 1.69/2.57 |
| Max Interval | 2.75/3.79 | 2.92/5.03 | 2.13/2.09 | 1.89/2.51 |
| MAE | 2.02/2.95 | 2.15/3.88 | 1.48/1.61 | 1.33/1.94 |
| MBE | −0.05/−0.41 | −0.08/−0.58 | 0.00/0.05 | 0.19/−0.21 |

Note: C is Calibration, V is Validation.

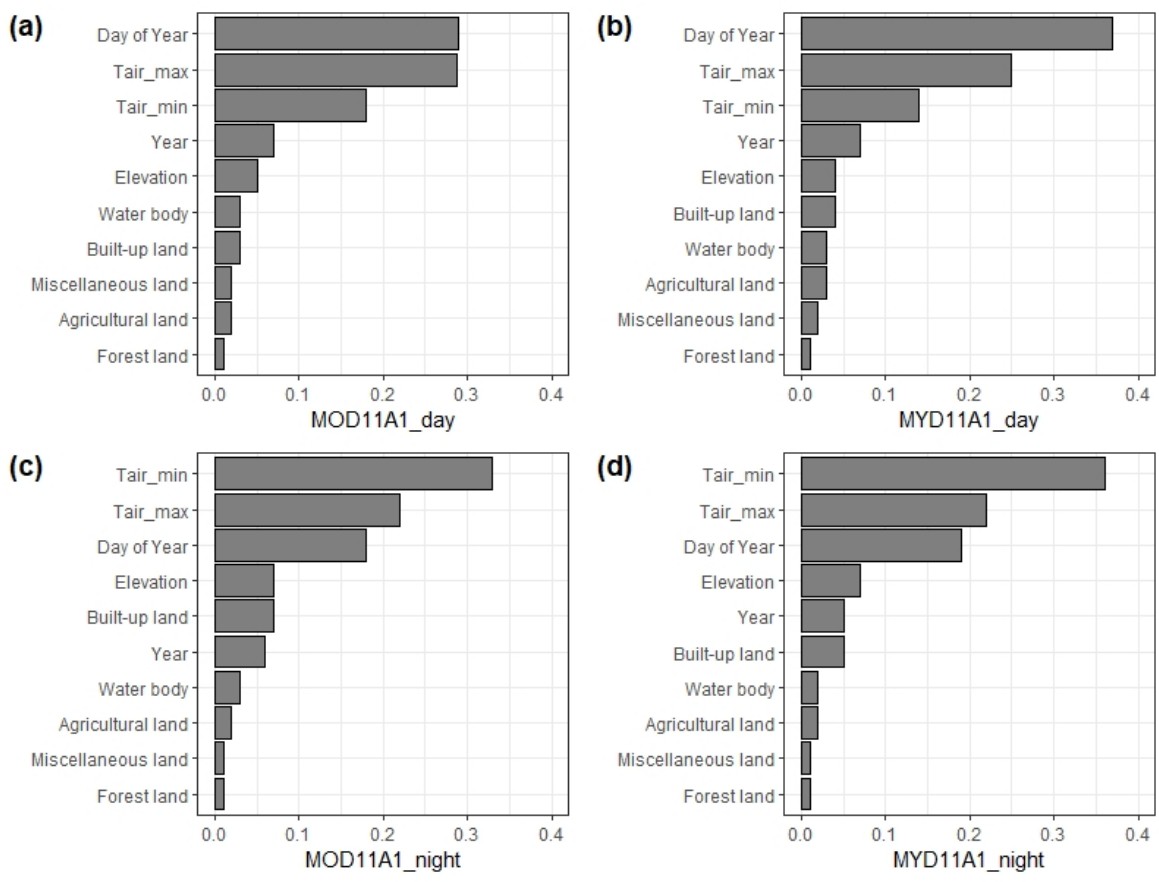

**Figure 3.** Importance of selected variables to predict LST for daytime (**a**,**b**) and nighttime (**c**,**d**).

### 3.2. Heatwave Detection and Measurement

3.2.1. Ground-Observed Air Temperature Heatwave

Our analysis of annual mean air temperature heatwave indices, considering both $T_{max}$ and $T_{min}$ heatwaves, reveals distinct patterns across different areas. In urban settings, the average yearly occurrence of daytime $T_{air}$ heatwaves ranges between 3.4 and 4.2 days,

compared to 5.2 to 6.8 days in peri-urban regions, and 3.6 to 4.0 days in rural locales. Notably, the highest frequency of HWF is observed in urban areas at 15.0 to 16.7 days per year, escalating to 26.4 to 27.1 days in peri-urban regions and 16.9 to 21.0 days in rural environments. The HWD extends to 4.5 to 5.0 days per year in urban centers, 5.5 to 6.0 days in peri-urban zones, and 5.9 to 6.5 days in rural areas. $T_{air}$ heatwave analysis reveals that the highest HWM in urban regions falls between 35.8 and 36.7 °C, 35.5 and 36.8 °C in the peri-urban area, and 34.3 and 37.6 °C in the rural area. The peak air temperatures of the HWA are recorded at 37.5–37.9 °C in urban areas, 37.6–38.1 °C in peri-urban zones, and 36.4–39.7 °C in rural settings. Consequently, the peri-urban area of Pathum Thani is distinguished by having the most significant occurrence of daytime heatwaves. Moreover, urban locales, especially Bangkok, persistently display the highest rates and lengths of heatwave events, and this trend has been progressively intensifying over the years. This observation highlights the critical need for focused climate adaptation strategies in these areas to mitigate the impacts of increasing heatwave activity.

Regarding nighttime $T_{air}$ heatwaves, the number of HWN in urban areas ranges from 3.0 to 3.5 days per year, paralleling the daytime patterns observed in other regions. Urban areas witness the maximum HWF, with 12.3 to 15.5 days per year, surpassing the 18.8 to 22.5 days in peri-urban regions and the 3.6 to 14.3 days in rural areas. For the longest HWD, it extends from 3.3 to 13.5 days per year in urban settings, 4.6 to 5.4 days in peri-urban areas, and 3.8 to 5.0 days in rural landscapes. The average cumulative HWM in urban locales is recorded between 35.8 and 36.7 °C, significantly higher than the 26.3–27.1 °C in peri-urban areas and 23.7–26.3 °C in rural settings. The peak air temperatures for the HWA reach 28.5–29.3 °C in urban areas, 27.8–28.8 °C in peri-urban zones, and 25.6–27.4 °C in rural environments. Thus, nighttime $T_{air}$ heatwave patterns exhibit similarities to their daytime counterparts, with urban regions being particularly prone to more frequent and severe events.

### 3.2.2. Satellite-Based Land Surface Temperature Heatwave

The result reveals the annual total number of daytime HWN, which ranges from two to nine. Pathum Thani emerges as the most affected region, followed by eastern Bangkok in terms of HWN prevalence. In terms of the annual HWF, which ranges from 10 to 39 days, the northern part of Pathum Thani exhibits the highest frequency. Regarding HWD, figures range from 4 to 11 days, with the rural regions of northern Saraburi experiencing the longest durations. The highest HWM during these events was recorded in the urban areas of the Don Muang region of Bangkok, with temperatures ranging from 33 °C to 42 °C. Additionally, the highest temperatures for the hottest HWA ranged from 35 °C to 45 °C, also observed in urban areas. To summarize, the annual distribution and characteristics of daytime heatwaves across different regions indicate that Pathum Thani and Eastern Bangkok are most prone to experiencing a higher number of HWN. Northern Pathum Thani endures the highest HWF of these events. Meanwhile, the rural areas of Northern Saraburi are subjected to the longest HWD. The urban regions, particularly in Don Muang, Bangkok, are notable for recording the highest HWM and temperatures on the hottest HWA. The data suggest that heatwaves are more frequent in peri-urban areas, while rural regions tend to experience longer-lasting heatwaves.

On the other hand, the analysis delineates the annual total number of nighttime HWN, which ranges from 3 to 12 events. Downtown Bangkok emerges as the area most affected by heatwave occurrences, closely followed by Northern Pathum Thani. The frequency of annual nighttime HWF varies from 13 to 62 days per year, with downtown Bangkok experiencing the highest frequency. Regarding HWD, the analysis shows a range from 4 to 12 days, with the longest observed in downtown Bangkok. The annual mean cumulative nighttime HWM spans from 21 °C to 27 °C, with the central districts of downtown Bangkok recording the highest values. The range of the warmest nighttime HWA extends from 21 °C to 28 °C, with the highest temperatures observed in central Bangkok. To conclude, the annual indices for nighttime heatwave events indicate that downtown Bangkok and

northern Pathum Thani are the most significantly affected regions, both in terms of the number of nighttime HWN and their HWF. Downtown Bangkok also records the longest HWD as well as the highest nighttime HWM and warmest nighttime HWA. The analysis suggests that nighttime heatwaves are more frequent and prolonged compared to their daytime counterparts, albeit with lower maximum temperatures. Urban areas, especially downtown Bangkok, are particularly vulnerable to severe nighttime heatwaves.

### 3.3. Spatial Homogeneity in Correlation Analysis

The degree to which the detected heatwave metrics from $T_{air}$ and LST align is demonstrated in Table 4. It presents the cumulative 'median' of Pearson's correlation coefficient ($r$) for the 10 meteorological stations and 10 LST grids that match the heatwave indices. Our LST modeling outperforms the existing MOD11A1 and MYD11A1 products during daytime heatwaves, notably for HWN at $r = 0.55, 0.62$, and HWF at $r = 0.66, 0.71$, respectively. It is evident that the derived in situ temperature largely follows the retrieved LST anomalies. For other indices, their associations are slightly moderate: HWD $r = 0.48$, HWM $r = 0.32$ to $0.39$, and HWA $r = 0.35$. In contrast to daytime heatwaves, nighttime heatwaves demonstrate a weaker link between all heatwave characteristics. It is evident that the derived in situ temperature largely follows the retrieved LST anomalies.

**Table 4.** Cumulative median Pearson's correlation coefficient ($r$) for linear relationship between 10 Observed $T_{air}$ points and 10 valid LST grids, coinciding with heatwave indices at $p < 0.05$.

| Data | Pearson's Correlation Coefficient ($r$) | | | | |
|---|---|---|---|---|---|
| | **HWN** | **HWF** | **HWD** | **HWM** | **HWA** |
| $T_{max}$ vs. MOD11A1 (day) | 0.55 | 0.66 | 0.48 | 0.32 | 0.35 |
| $T_{max}$ vs. MYD11A1 (day) | 0.62 | 0.71 | 0.48 | 0.39 | 0.40 |
| $T_{min}$ vs. MOD11A1 (night) | 0.31 | 0.26 | 0.10 | 0.02 | 0.07 |
| $T_{min}$ vs. MYD11A1 (night) | 0.36 | 0.45 | 0.26 | 0.08 | 0.13 |

Figure 4 illustrates the distribution of the correlation coefficient (Pearson's $r$) at $p < 0.05$ corresponding to heatwave indices, considering pixel-by-pixel observations of $T_{air}$ and LST. In rural, peri-urban, and urban areas, the link between $T_{max}$ and MOD11A1 and $T_{max}$ and MYD11A1 is positively strong, with $r$ values larger than 0.60 and up to 0.90, respectively; the highest correlated MOD11A1 result is observed in P07 (located in Ayutthaya province as a rural context). In addition, the association between $T_{max}$ and MYD11A1, particularly in rural regions in Lopburi province, P08 and P09, remains more robust than in other places ($r = 0.80$) for HWN, HWF, and HWD. At $r = 0.60$, however, peri-urban and urban stations are strongly related to HWM and HWA. In a peri-urban area, the connection between nighttime heatwaves corresponding to $T_{min}$ and MOD11A1 is highest for HWN ($r = 0.85$) and HWF ($r = 0.75$) in Suvarnabhumi Airport (P05). In contrast, HWD shows the strongest link in the urban region (P03 Bang Na, Bangkok) with a correlation coefficient of 0.75, while P01 (Khlong Toei, Bangkok) displays the highest HWM ($r = 0.65$) and HWA ($r = 0.62$). The $T_{min}$ against the MYD11A1 product indicates that P05, as the outskirt area, has the strongest link with all heatwave indices at $r = 0.60$ to $0.85$, with the exception of P02 as the urban area, which has a smaller magnitude. Nighttime heatwaves are discovered to have a negative correlation in more regions than daytime heatwaves, particularly in rural areas.

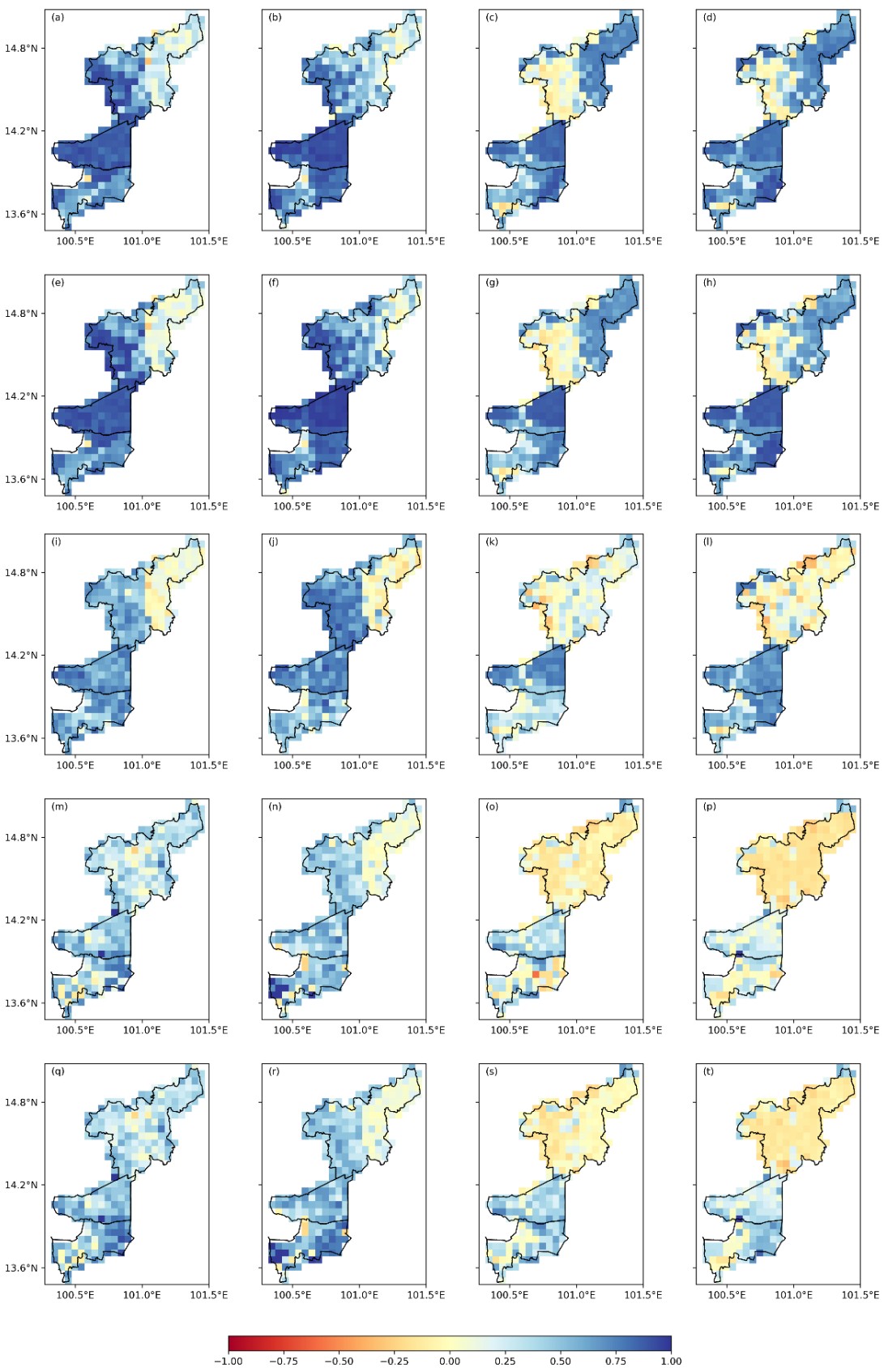

**Figure 4.** The distribution of pixel-wise correlation coefficients (*r*) between observed $T_{\text{air}}$ and observed MODIS-LST; $T_{\text{max}}$ and MOD11A1 Day (**a,e,i,m,q**), $T_{\text{max}}$ and MYD11A1 Day (**b,f,j,n,r**), $T_{\text{min}}$ and MOD11A1 Night (**c,g,k,o,s**), $T_{\text{min}}$ and MYD11A1 Night (**d,h,l,p,t**).

## 4. Discussion

### 4.1. Performance Evaluation of Land Surface Temperature Predictive Modeling

Given the inherent limitations of MODIS-LST, which often contains gaps due to cloud cover and other atmospheric conditions [47,69], filling these missing data points is particularly critical in heatwave assessment studies that require daily data inputs rather than instantaneous or averaged values. Consequently, a crucial step in heatwave assessment involves the optimization of the dataset during preprocessing, which includes a systematic treatment of missing values essential for accurate heatwave detection. Selecting appropriate input variables is crucial before training machine learning models [112]. Our results align with the findings of [113], confirming that LST patterns are not constant and exhibit seasonal variations. By selecting a more varied set of predictors than those used in previous studies, our study highlights the critical importance of temporal factors in LST analysis. It significantly enhances heatwave analysis by exploring the implications of identified correlations between different variables, particularly in understanding the dynamics of heatwaves and their impacts.

Our selected methodology, RF, aligns with the findings of [114], which underscore the importance of incorporating a diverse range of environmental and land-use factors in predicting LST. By employing both temporal and spatial variables as significant predictors for LST modeling, our approach not only advances our understanding of LST but also highlights the pronounced impact of temporal variables. This emphasis suggests that these factors may have a more substantial effect on LST than previous studies. Moreover, our work builds upon the findings of [47,115,116], which linked LST with factors such as land cover types, terrain, vegetation, moisture conditions, solar radiation, elevation, and Julian day. Interestingly, our research has distinctively identified the DOY as the most significant feature for predicting daytime LST (MOD11A1; $R^2$ = 0.28 and MYD11A1; $R^2$ = 0.36) as shown in Figure 3, owing to its critical role in influencing weather patterns through its capture of seasonal changes. Similarly, the daily $T_{min}$ is crucial for nighttime LST prediction (MOD11A1; $R^2$ = 0.33 and MYD11A1; $R^2$ = 0.36), confirmed by the study of [117], which stated that the average of nighttime LST was closest to $T_{min}$, addressing diurnal temperature variations in both urban and non-urban settings.

The outstanding performance of the MOD11A1 night model is supported by its RMSE of 2.09 °C and high $R^2$ value of 0.64. These results align with theories suggesting that certain algorithms possess superior capabilities in processing and analyzing complex environmental data, particularly during stable atmospheric conditions at night. Our results are consistent with those of [62], who demonstrated the high spatial–temporal continuity and relative accuracy of reconstructed nighttime MODIS-LST products across China. However, temporal feature extraction, which involves incorporating time-related variables such as day of the year to account for seasonal effects on temperature, is significant. This process is reflected in the challenges faced by the MYD11A1 day model, which showed data overestimation issues with the highest RMSE (5.02 °C) and the lowest $R^2$ (0.29). In contrast with the study of [118], it was found that MOD11A1 and MYD11A1 products slightly underestimated daytime LST with an overall absolute bias < 0.9 °C and RMSE < 2.9 °C. The effective use of the MOD11A1 night model and the challenges associated with the MYD11A1 day model are consistent with theories in environmental data analysis. This demonstrates a nuanced understanding of how algorithms perform under various atmospheric conditions. However, when utilizing LST for heatwave studies, the requirement for daily data is critical and should be a primary consideration in future research.

The issue of data overestimation in the MYD11A1 day model highlights the practical challenges encountered when applying theoretical models. This phenomenon can be attributed to the impact of solar radiation on the thermal infrared signal during daytime, thereby introducing complexity to the relationship between $T_{air}$ and LST [116]. Ref. [38] suggested that utilizing both Terra and Aqua satellites enabled the development of consistent indices suitable for analysis during both day and night. This methodology revealed that anomalies in LST effectively mirror the climatic trends of the area and reliably sig-

nal the occurrence of heatwaves, especially the more intense ones, which can be further employed in future studies. Adjusting hyper-parameters in the RF model can lead to overfitting, particularly when the rules are overly complex and tailored to the training data. Employing unseen test data and cross-validation methods can help mitigate this risk and ensure improved performance on new datasets [119,120]. Furthermore, regarding the use of the Normalized Difference Vegetation Index (NDVI) and the Normalized Difference Built-Up Index (NDBI) for downscaling the TIR bands, an improvement in performance accuracy should be further considered in modeling LST [121,122].

### 4.2. Heatwave Detection, Its Magnitude, and Characteristics

#### 4.2.1. Ground-Observed Air Temperature Assessment

In examining the spatial and temporal patterns of heatwaves, our findings highlight a distinct trend across various regions. During daytime heatwaves, peri-urban areas such as Pathum Thani and urban locations like Don Muang in Bangkok are particularly affected, showing significant increases in HWF, HWD, HWM, and HWA. This phenomenon is largely attributed to the urban environment, characterized by construction materials such as concrete, asphalt, and steel, and unique morphological features that inherently possess heat retention properties. These factors contribute to longer, more frequent, and intense heatwave conditions [123]. The study of [124] found that the cumulative hours of extreme heat waves increased significantly with the proportion of urban land and decreased significantly with the proportion of forested land and water. Additionally, the analysis highlights the years 1997, 2016, and 2019 as having the highest instances of each heatwave index, indicating an upward trend in the occurrence of heatwave events. The peri-urban station in Pathum Thani showed the most significant increases in daytime HWN, HWF, and HWD, while the most pronounced nighttime HWN increases occurred in rural areas near Saraburi. This pattern highlights the growing susceptibility of peri-urban areas to heatwaves, exacerbated by rapid urbanization, diminishing green spaces, and local climate factors that enhance heatwave sensitivity. This observation is consistent with the research of [30,125–132], which associated severe heatwaves with the strongest El Niño–Southern Oscillation (El Niño) years on record, including 1998, 2010, and 2016. In opposition to nighttime heatwave, examining heatwave detection through $T_{\min}$ unveils unique patterns and trends distinct from those associated with $T_{\max}$. Notably, urban areas such as Bang Na, Klong Toei, and Don Muang in Bangkok exhibit the most pronounced heatwave activities, particularly in terms of frequency and duration, while peri-urban regions like Pathum Thani also experience a high incidence of events. The years 2013 and 2019 are marked as having significant impacts, with the highest values recorded for various indices of nighttime heatwaves. Across both day and night, an increase in heatwave metrics was observed in 1997, 2013, 2016, and 2019, coinciding with severe global heatwaves during the most intense El Niño events recorded so far regarding to [133–136]. These observations align with broader research efforts in the field, such as those by [26,137], which note significant climatic shifts impacting heatwave patterns, particularly in urban settings. The findings underscore the critical importance of understanding regional heatwave trends and their interactions with biophysical changes, human activities, and land use shifts. The results of this study are particularly concerning in light of the ongoing expansion of urban areas into rural landscapes, which could further exacerbate heatwave conditions.

#### 4.2.2. Satellite-Based Land Surface Temperature Assessment

Comparative analysis using satellite data alongside air temperature measurements further confirms the notable difference, with satellite observations specifically highlighting the enhanced intensity of nighttime heatwaves in comparison to daytime events across various metrics. This observation aligns with the findings of [106], which suggest that nighttime and compound heatwaves experience a more pronounced increase in both frequency and intensity compared to their daytime counterparts. Notably, peri-urban and rural regions display elevated heatwave metrics consistently. Further, the examination of

annual heatwave patterns at specific sites reveals that daytime heatwave events are more prevalent and severe in peri-urban areas, such as Pathum Thani and Eastern Bangkok, with occurrences ranging from 2 to 9 and durations from 10 to 39 days within a year. In contrast, urban areas, specifically downtown Bangkok, face a higher risk of nighttime heatwaves, with an annual frequency of 3 to 12 events and durations ranging from 13 to 62 days. Our findings reveal a significant rise in nighttime heatwave number (HWN) and heatwave amplitude (HWA) trends in urban settings, particularly in the Bang Na and Suan Luang districts of Eastern Bangkok. These trends suggest a potential amplification of the urban heat island effect, likely influenced by rapid urban development. The construction of new residential projects, warehouses, and buildings, alongside proximity to industrial estates and Suvarnabhumi International Airport, may contribute to this phenomenon. A study by [138] supports this correlation, indicating that a 10% increase in urban built-up density can lead to a 0.08% to 0.95% rise in HWN. Our study highlights that LST demonstrates lower amplification for HWM while being higher for HWA compared to air temperature measurements, particularly in urban settings during daytime heatwaves. Our findings, in line with [37], reveal that comparing air temperatures and satellite-derived LST data between normal and heatwave years shows a significant increase in daytime air temperature during heatwaves. For example, the Don Muang region in Bangkok exhibited the highest annual temperature extremes, with observed HWM fluctuating between 33 °C and 42 °C, and HWA extending from 35 °C to 45 °C. Significantly, the highest mean HWM of $T_{air}$ was observed in UBKK (Khlong Toei, Bangkok) at 45.5 °C in 1997, and the highest HWA was recorded in RBCL (Bua Chum Meteorological Station, Lop Buri) at 43.2 °C in 2016. Our findings contrast with the reported effectiveness of LST data in identifying daytime heatwaves as underscored by [38,139], demonstrating the strong correlations between $T_{air}$ and MODIS-LST data. However, ref. [140] demonstrates how the suggested GIS-based methodology may be used to analyze heatwave susceptibility and effect scenarios in different urban patterns. This result underscores the need for targeted climate adaptation and resilience strategies, particularly in urban and peri-urban areas, to address the distinct impacts of daytime and nighttime heatwaves. Therefore, these findings are crucial for understanding heatwave dynamics, providing essential insights for anticipating and mitigating heatwave impacts, which are increasing in frequency and severity due to climate change.

*4.3. Spatial Homogeneity in Correlation Analysis between Detected Heatwave Indices from Ground-Observed Air Temperature and Satellite-Based Land Surface Temperature*

Spatial–temporal consistency evaluates the uniformity of identified heatwave patterns across varied regions and temporal spans. This evaluation is frequently quantified through the use of correlation coefficients or measures of similarity. By conducting comparisons of identified heatwave events with ground truth data and evaluating the detection algorithms' reliability, the study can affirm the integrity of the findings. In the context of satellite-derived LST for heatwave detection, metrics such as detection accuracy, false alarm rate, and spatial–temporal consistency are scrutinized [141]. Nonetheless, it is imperative to consider additional factors, including spatial coverage, resolution, and the spectral properties of the satellite data, to ensure a comprehensive assessment.

Our investigation, employing Pearson's correlation analysis, reveals a significant relationship between observed heatwaves based on $T_{air}$ and MODIS-derived LST across various areas, particularly in its correlation with heatwave indices as detailed in Table 4. This dual approach is comprehensive, as it covers both the atmospheric temperature (felt by residents) and the surface temperature (which influences the local microclimate). During daytime heatwave conditions, the findings are noteworthy, especially in the context of two specific heatwave characteristics, HWN and HWF, with $r = 0.55$–$0.71$ indicating a strong positive correlation. Other aspects, such as HWD, HWM, and HWA, demonstrate a moderate association. Consistent with the findings of [139], the observed match percentages are relatively high, especially when considering the differences in terms of HWM. Our

results contrast daytime and nighttime heatwave conditions, noting weaker correlations across all heatwave characteristics during the nighttime, with $r = 0.02$–$0.45$.

In particular, the degree that determines the spatial–temporal pairwise correlation is initially measured as shown in Figure 4. According to the findings of our research, grids located in rural areas have the potential to form the strongest associations compared to the other grids in terms of the HWN, HWF, and HWD, as measured by $r = 0.93$, $r = 0.94$, and $r = 0.80$, respectively. On the other hand, the correlation coefficients for HWM and HWA are found to be at their maximum in a peri-urban area (Pathum Tani), with values of $r = 0.65$ and $r = 0.85$, respectively. Overall, this study represents a significant step forward in our ability to model and understand heatwaves through LST data, especially during daytime heatwaves, and it is crucial, as it highlights the challenges in modeling nighttime heatwaves. This agrees with the findings of [38], which showed that the proposed LST index effectively identified heatwaves in the Mediterranean region during the daytime; however, this correlation was slightly weaker during the nighttime.

The utilization of MODIS-retrieved LST datasets for heatwave mapping, especially in peri-urban and rural regions with limited meteorological data, marks a significant advancement over traditional methods. This is particularly beneficial, as it offers the potential for repetitive imaging at frequent sampling times [38]. The strong correlations in rural and peri-urban areas, coupled with the varied correlations in urban settings, underscore the intricate interplay between environmental factors and heatwave dynamics across diverse geographical landscapes. Our approach not only allows for comprehensive mapping of areas prone to heatwaves but also emphasizes the importance of extensive geographic analysis and demographic information in understanding heatwave patterns. Moreover, ref. [142] confirmed that both surface imperviousness (SI) and LST could be used to better understand spatial variation in heat exposures over longer time frames but are less useful for estimating shorter-term, actual temperature exposures, which can be useful for public health preparedness during extreme heat events.

### 4.4. Summary of Comparative Analysis with Existing Related Works

Our LST predictions are consistent with the conclusions drawn in various related studies that utilize the RF model and consider both temporal and spatial variables (refer to [49,112,113,143]). These studies present compelling findings akin to ours. Moreover, our methodology, which emphasizes a wide array of predictors, especially temporal factors, not only corresponds to but also advances upon prior research (refer to [47,114–117]). Notably, our analysis identifies the Day of Year (DOY) and minimum daily air temperature ($T_{min}$) as pivotal in predicting daytime and nighttime LST, respectively. However, this finding contrasts with [118], which noted a slight underestimation in LST predictions using MOD11A1 and MYD11A1 products. Thus, our work highlights the nuanced contributions of diverse environmental, land-use, and temporal variables to accurate LST prediction, thereby enhancing model performance and reliability.

In agreement with [123], our study highlights the significant impact of $T_{air}$ heatwaves in regions such as Pathum Thani (suburban) and Bangkok (urban), resulting in these areas being the most susceptible to global warming. We reveal the link between increased heatwave metrics (HWF, HWD, HWM, HWA) and significant El Niño events in 1997, 2013, 2016, and 2019, a finding supported by [133–136]. Our results also align with the research by [26,137], providing further evidence of how climatic shifts influence urban heatwave patterns, thereby enriching our understanding with a comprehensive, referenced analysis. Moreover, comparing satellite and in situ air temperature data revealed more intense nighttime heatwaves, consistent with [106], resulting in a pronounced increase in nighttime heatwave frequency and intensity, which relates to the UHI effect. Contrasting with [139] on the use of LST data, our findings reveal a significant correlation between air temperature and MODIS-LST data with robust daytime correlations between Tair and LST in rural (HWN, HWF, HWD, $r > 0.90$) and peri-urban (HWM, HWA, $r > 0.65$) regions, challenging traditional perspectives and highlighting the complexity of heatwave analysis.

Furthermore, our integration approach unveils key insights into daytime heatwave dynamics, with strong correlations in heatwave number and frequency, and moderate connections in duration, magnitude, and amplitude, both supporting and diverging from the findings of [139]. Specifically, rural areas demonstrate pronounced spatial–temporal correlations, highlighting the efficacy of our method in detecting heatwave patterns through LST data—an essential advancement for tackling the challenges of nighttime heatwave analysis, as noted by [38]. By utilizing MODIS-derived LST datasets, our study enhances the understanding of heatwave episodes in less monitored peri-urban and rural areas, significantly improving upon traditional methods. This approach, validated by [142], emphasizes the importance of surface imperviousness and LST in mapping the spatial variability of heat exposure, underlining the intricate interplay between environmental factors and heatwave dynamics.

Ultimately, this study effectively utilizes the RF model along with temporal and spatial predictors to enhance LST predictions, providing critical insights into the dynamics of daytime and nighttime heatwaves. Our findings highlight the vulnerability of urban and peri-urban areas like Bangkok and Pathum Thani to heatwaves, exacerbated by climatic shifts such as significant El Niño events and global boiling effects. By deepening our understanding of these impacts, our research supports the development of informed climate adaptation strategies, crucial for improving urban and non-urban resilience against escalating heatwave frequencies and intensities.

## 5. Conclusions, Perspectives, and Possible Future Works

This study addresses a critical knowledge gap concerning the challenges posed by heatwaves in Thailand, particularly emphasizing the need for comparative analyses across urban, peri-urban, and rural settings. By integrating geospatial analysis, remote sensing, and terrestrial data, our innovative methodology meticulously maps heatwave patterns across three socio-economic regions: Bangkok (urban), Pathum Thani (peri-urban), and Saraburi (rural). The integration of satellite LST data with ground-based observations has provided a nuanced understanding of heatwave patterns, emphasizing the complex dynamics that influence urban planning and public health. The methodology introduced in this study is distinctive in its comprehensive approach to combining various data types and analytical techniques to improve the precision of heatwave predictions.

Our approach enhances the understanding of the spatial and temporal dynamics of heatwaves, providing the most detailed collection of air temperature and satellite-derived heatwave data to date. We effectively utilize the RF model alongside a diverse array of both temporal and spatial predictors to refine LST predictions, aligning with and advancing existing research. Key variables, such as the Day of Year and minimum daily air temperature, have been identified as essential for accurately predicting daytime and nighttime LST. This robust, accurate, and reliable methodology not only advances current practices but also demonstrates the potential of machine learning techniques, particularly the MODIS-MOD11A1 and MODIS-MYD11A1 models, in environmental data analysis despite challenges such as data gaps and the need for ongoing model refinement.

Our study shows that both air temperature and satellite-derived LST data reveal significant vulnerabilities to heatwaves in urban and peri-urban areas, particularly in places like peri-urban Pathum Thani and urban Bangkok, which suffer from prolonged heatwave durations. We find that these areas, especially downtown Bangkok and Northern Pathum Thani, experience increasingly frequent and severe heatwaves, predominantly at night. A unique aspect of our research is the demonstration that LST can reliably serve as a proxy for heatwave analysis due to its high spatial correlation with ground-based measurements. This capability is particularly valuable in regions without weather stations. Highlighting the urgent need for effective climate adaptation strategies, our study leverages satellite LST data to enhance our understanding and response to heatwave impacts in densely populated regions where traditional data sources are unavailable.

Furthermore, the limitations posed by the data, including the misalignment of temporal scales between the air temperature and MODIS-derived land surface temperatures, as well as gaps in the observational records, were significant. Despite these challenges, the study employed the most accessible datasets available, illustrating the need for improved data collection methods to enhance the reliability of future predictions. Looking ahead, future research should focus on developing comprehensive and interdisciplinary strategies to mitigate the impacts of heatwaves. Building on the foundation laid by this study, further efforts could explore the integration of public health, urban planning, and community resilience frameworks to develop holistic solutions. Additionally, the employment of advanced ensemble methods, such as artificial neural networks (ANNs), gene expression programming (GEP), and gradient boosting models (GBMs), could enhance the accuracy and reliability of predictions.

In conclusion, this comprehensive study not only corroborates previous findings but also introduces new insights into the spatial–temporal dynamics of heatwaves, thereby enriching the scientific dialogue on urban and peri-urban climate resilience. The findings lay the groundwork for targeted climate adaptation strategies, enabling urban planners and policymakers to more effectively mitigate the adverse effects of heatwaves. Our contributions to understanding climate resilience are significant, revealing the severe impacts of heatwaves through a methodological framework that is applicable both in Thailand and globally. The insights provided here highlight the urgent need for continuous research and the development of effective adaptation and mitigation strategies to protect communities from the escalating severity of climate-related challenges. Moreover, the advancements detailed in this study are vital for informing government policies and decision-making processes, enhancing resilience against climate extremes in both urban and non-urban settings, and driving forward efforts towards a sustainable future.

**Author Contributions:** Conceptualization, T.C., A.T. and H.M.; methodology, T.C., A.T. and H.M.; software, T.C.; validation, T.C.; formal analysis, T.C.; investigation, T.C., A.T. and H.M.; resources, T.C.; data curation, T.C.; writing—original draft preparation, T.C.; writing—review and editing, A.T., H.M. and T.W.T.; visualization, T.C.; supervision, A.T., H.M. and T.W.T.; funding acquisition, T.C. All authors have read and agreed to the published version of the manuscript. The authors confirm the copyright of the figures and tables in the manuscript.

**Funding:** This research received no external funding.

**Institutional Review Board Statement:** Not applicable.

**Informed Consent Statement:** Not applicable.

**Data Availability Statement:** The raw data supporting the conclusions of this article will be made available by the authors on request.

**Acknowledgments:** The authors would like to express their deep appreciation to the Land Development Department and the Thai Meteorological Department for their invaluable contributions to land use/land cover and topographic data, and meteorological data, respectively. Acknowledgments are also extended to Abhishek Koirala for his contributions to the development of scripts for heatwave detection and to Siwat Kongwarakom for his support in coding the random forest-based land surface temperature imputation. Finally, the authors also wish to thank the reviewers for their thorough review of our manuscript and for providing valuable suggestions.

**Conflicts of Interest:** The authors declare no conflicts of interest.

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
