# Peer review of "Integrating Remote Sensing and Ground-Based Data for Enhanced Spatial–Temporal Analysis of Heatwaves: A Machine Learning Approach"

_applsci, doi:10.3390/app14103969_

Round 1
Reviewer 1 Report
Comments and Suggestions for Authors
In my opinion the paper is very well written, and the research is very well organized. The reader can compare and understand all the spatiotemporal differences in MODIS derived part, but overall is fine. I am very satisfied to see that the authors deal with data gaps satisfactory. This is a very good approach to studying heatwaves in Asia and I think it is worthy of publication.
I would only suggest rephrasing Lines 622-626 and make syntax better.
Thank you
Author Response
Dear Reviewer,
Your feedback is invaluable to us, and we are grateful for the time and effort you have invested in reviewing our work.
Please see the attachment.
Sincerely yours,
Thitimar Chongtaku*, Attaphongse Taparugssanagorn*, Hiroyuki Miyazaki**, and Takuji W Tsusaka***
* School of Engineering and Technology, Asian Institute of Technology, Thailand
** Center for Spatial Information Science, University of Tokyo, Japan
*** School of Environment, Resources and Development, Asian Institute of Technology, Thailand

Reviewer 2 Report
Comments and Suggestions for Authors
The subject of interest of this paper is the dynamic of prolonged periods of excessively hot weather – the so-called heatwaves. The study area includes the urban, peri-urban, and rural regions within the central area of Thailand. As the contribution, it proposes a novel approach to understanding heatwave dynamics, which combines satellite-derived land surface temperature data with in-situ air temperatures and some other spatial and temporal features. The manuscript is well written, and it contains all required elements including a separate section explaining in detail the contribution, an excessive discussion where the obtained results are interpreted in perspective of previous studies, as well as future research directions highlighted in the conclusions section. I have not any suggestion.
Author Response

(The authors gave the same response as above.)

Reviewer 3 Report
Comments and Suggestions for Authors
Dear and respective authors,
The manuscript entitled "Integrating Remote Sensing and Ground-Based Data for Enhanced Spatiotemporal Analysis of Heatwaves: A Machine Learning Approach" presents a promising methodology for advanced understanding of heatwaves dynamics and improvement of heatwave management practices. The integration of remote sensing and ground-based data, coupled with a machine learning approach, offers a comprehensive and innovative solution to analyze heatwave patterns with improved accuracy. This was the main question addressed by the research, respectively. The manuscript showcases thorough research and demonstrates potential applications in the field related to climatic hazards analysis. Additionally, it contributes to the study on heatwave detection and monitoring in Southeast Asia (with Thailand as a case study). The methodological section is well conceptualized, depicting the novel approaches that this study nurtures. Furthermore, the conclusions are consistent with the evidence and arguments presented in this research. With minor revisions, this study has the potential to make a valuable contribution to the Applied Sciences journal. Below are the listed minor comments that should be addressed in order to enhance the overall quality of the manuscript:
1. Change "Spatiotemporal" to "Spatial–Temporal" in the manuscript title.
2. In the introduction (line 26, page 1) add sentence and reference: "Over the initial two decades of the 21st century, a multitude of severe heatwaves had a profound impact on human health, agriculture, water resources, energy demand, regional economies, and forest ecosystems (Basarin et al., 2020)." insted of stentence "They pose a persistent and severe threat to human well-being and the environment." * Basarin, B.; Lukić, T.; Matzarakis, A. Review of Biometeorology of Heatwaves and Warm Extremes in Europe. Atmosphere 2020, 11, 1276. https://doi.org/10.3390/atmos11121276.
3. Please elaborate more on the heatwave related problems in the South East Asia region. The authors can cosult some partient literature such as: * Huang, C.; Cheng, J.; Phung, D.; Tawatsupa, B.; Hu, W.; Xu, Z. Mortality Burden Attributable to Heatwaves in Thailand: A Systematic Assessment Incorporating Evidence-Based Lag Structure. Environment International 2018, 121, 41–50, doi:10.1016/j.envint.2018.08.058. * Lee, H.; Punnasiri, K.; Tong, S. Effects of Temperature on Mortality in Chiang Mai City, Thailand: A Time Series Study. Environmental Health 2012, 11, doi:10.1186/1476-069x-11-36. * Shrestha, R.P.; Chaweewan, N.; Arunyawat, S. Adaptation to Climate Change by Rural Ethnic Communities of Northern Thailand. Climate 2017, 5, 57. https://doi.org/10.3390/cli5030057 * Dong, Z.; Wang, L.; Sun, Y.; Hu, T.; Limsakul, A.; Singhruck, P.; Pimonsree, S. Heatwaves in Southeast Asia and Their Changes in a Warmer World. Earth’s Future 2021, 9, doi:10.1029/2021ef001992.
4. In the Introduction section of the literature review, please add a couple of sentences regarding current heatwave management practices in Thailand. The authors are advised to cosult the work of * Arifwidodo, S.D.; Chandrasiri, O. Urban Heat Stress and Human Health in Bangkok, Thailand. Environmental Research 2020, 185, 109398, doi:10.1016/j.envres.2020.109398.
5. Throughout the text, please use the term "air temperature" instead of "temperature."
6. In the Table 2. please insert an additional column with references related to the definitions of the used heatwave indices. 7. After sub-section 2.3.3, insert a short description of the software used for the graphical representation and mapping of the study-derived results. This sub-section can be labeled as 2.3.4. 8. Figure 3. Has low quality. Please enhance it. 9. Page 16, line 517: please modify the sub-title 4.2 into "Heatwave detection, its magnitude, and characteristics".
10. Please exclude the reference [95] (landslide mapping) from the research since it does not fit the topic of the manuscript.
11. Page 18, lines 642-644: please indicate the correlation value between air temperature and MODIS-LST data
12. Reference list should be enhanced a bit (according to the previous comments).
Minor editing of English language required.
Author Response

(The authors gave the same response as above.)

Reviewer 4 Report
Comments and Suggestions for Authors
Dear authors, this study was really interesting to read and to learn something new about this part of the world. Please find below a few suggestions on how to improve your work:
Abstract:
In the abstract, author mention that that heatwaves are underreported and under-researched in Thailand, please add a sentence why addressing this gap is important.
Authors mention the main heatwave metrics analyzed and present the key findings regarding heatwave patterns. Nevertheless, the abstract could be improved by clearly stating the most significant findings or trends observed, whether were there any notable increases or changes in heatwave intensity, frequency, etc., over the study period.
Introduction:
Introduction could benefit from clearer segmentation or subsections to make it easier for the reader to go through different themes. For example, the transition from global heatwave trends to Thailand's situation could be smoother.
Even thou, the author mentions the impacts of heatwaves, consider adding more specific examples or statistics to illustrate these impacts. You could quantify the effects on health (number of deaths for example during one heatwave), infrastructure, agriculture (losses in millions of $), and the economy would make the consequences more understandable for readers.
Please follow the inverted pyramid structure, start from a general theme, and then gradually narrow it down to focus on your problem.
Research gap
Why is subsection 1.1 bolded and 1.2 is not? I assume it is a mistake, please correct it.
Materials and Methods
In the study area description, row 152 you mention: “These classifications are based on national standards,…” What are those standards? Please provide a short description. Or you are thinking about the bullet points “Urban”, and “Peri-urban”… it is not clear.
Figure 1, (a) longitude and latitude are not easily readable, please enlarge it.
Figure 1 (b) in Legend, “High 1113” is really hard to observe, please enlarge it.
Consider providing a short explanation of the generalized split-window (GWS) algorithm that you used for retrieving MODIS-LST. This could be helpful for readers to understand the method behind the temperature measurements.
Mention potential limitations associated with MODIS-LST data, e.g., spatial or temporal resolution limitations, cloud cover issues, etc.
Results
Figure 3, titles on x-axis and y-axis are a little fuzzy, not clear for readers.
Discussion
Each section within the discussion nicely explains the findings and compares them with existing and up-to-date scientific literature. However, try to make smoother transitions between sections this can improve the overall flow and readability of the discussion.
Consider highlighting how your results contribute to advancing knowledge in the field. More specifically, what is the novelty or significance of your findings compared to previous research, this can strengthen the discussion.
Please provide deeper interpretation and insights of the observed trends and correlations e.g., discuss what are the potential implications of the identified correlations between different variables for understanding heatwave dynamics and their impacts.
Try to include a brief summary at the end of the discussion only to recap the key findings and their implications for advancing knowledge in the field of heatwave research. This can serve nicely as an introduction to the Conclusion section.
Conclusion
Clearly highlight the most important results of the study and how they contribute to existing knowledge.
Consider explaining the novelty or uniqueness of the methodology used in the study.
End the conclusion with a concise summary that reiterates the significance of the study's findings and highlights the importance of continued research efforts in mitigating the impacts of heatwaves.
Author Response

(The authors gave the same response as above.)

Reviewer 5 Report
Comments and Suggestions for Authors
Dear authors, please find my comments and suggestions in the attached file.

Author Response

(The authors gave the same response as above.)

Round 2
Reviewer 5 Report
Comments and Suggestions for Authors
Dear authors, thank you for the interesting research!